# Effect of Roselle Flower Extract (*Hibiscus sabdariffa* Linn.) on Reducing Steatosis and Steatohepatitis in Vitamin B12 Deficiency Rat Model

**DOI:** 10.3390/medicina59061044

**Published:** 2023-05-28

**Authors:** Irena Ujianti, Imelda Rosalyn Sianipar, Ani Retno Prijanti, Irsan Hasan, Wawaimuli Arozal, Ahmad Aulia Jusuf, Heri Wibowo, Joedo Prihartono, Patwa Amani, Dewi Irawati Soeria Santoso

**Affiliations:** 1Departement of Medical Physiology, Faculty of Medicine, Universitas Muhammadiyah Prof. Dr. Hamka, Jakarta 121130, Indonesia; 2Graduate Student in Doctoral Program in Medical Sciences, Faculty of Medicine Universitas Indonesia, Jakarta 10430, Indonesia; 3Departement of Medical Physiology, Faculty of Medicine, Universitas Indonesia, Jakarta 10430, Indonesia; 4Departement of Biochemistry and Molecular Biology, Universitas Indonesia, Jakarta 10430, Indonesia; 5Departement of Internal Medicine, Universitas Indonesia, Jakarta 10430, Indonesia; 6Departement of Pharmacology and Therapeutic, Universitas Indonesia, Jakarta 10430, Indonesia; 7Departement of Histology, Universitas Indonesia, Jakarta 10430, Indonesia; 8Departement of Parasitology, Universitas Indonesia, Jakarta 10430, Indonesia; 9Departement of Community Medicine, Universitas Indonesia, Jakarta 10320, Indonesia; 10Departement of Medical Physiology, Faculty of Medicine, Universitas Trisakti, Jakarta 11440, Indonesia

**Keywords:** inflammation, lipogenesis, liver, roselle, vitamin B12, steatosis, steatohepatitis

## Abstract

*Background and Objectives*: Non-alcoholic Fatty Liver Disease (NAFLD) can occur as a result of micronutrient deficiencies. Hibiscus sabdarifa, a plant used in traditional medicine, contains ingredients that can help prevent this process. This study looked at the potency of *Hibiscus sabdariffa* Ethanol Extract (HSE) to prevent homocysteine-induced liver damage in animals that were deficient in vitamin B12. *Materials and Methods*: A comparative study of the effects of roselle extract is presented in an experimental design. Thirty Sprague–Dawley rats were divided into six groups using randomization. To demonstrate the absence of liver damage in the experimental animals under normal conditions, a control group was fed a normal diet without HSE. For the induction of liver damage in the experimental animals, the vitamin B12-restricted group was administered a vitamin B12-restricted diet. To test the effect of HSE on liver damage, the treatment group was given HSE along with a vitamin B12-restricted diet. Each group was given two treatment periods of eight and sixteen weeks. These results were compared with the results of the parameter examination between the vitamin B12 restriction group, with and without HSE, using an ANOVA statistic. The data were analyzed with licensed SPSS 20.0 software. *Results*: HSE significantly increased the blood levels of vitamin B12 while lowering homocysteine levels. The administration of HSE reduced liver damage based on the activity of liver function enzymes in the plasma due to a limitation of vitamin B12. HSE decreased Sterol Regulatory Element-Binding Protein-1c (SREBP1c) and Nuclear Factor Kappa B (NFkB) protein expressions in the liver tissue, but did not decrease Glucose-Regulated Protein 78 (GRP78) protein expression. Significantly, the levels of Tumor Necrosis Factor alpha (TNF-a) and IL-6 in the liver tissue were lower, while the levels of IL-10 and Nuclear factor-erythroid-2 Related Factor 2 (NRF2) were higher with HSE administration. HSE produced a better histopathological profile of the Hematoxylin and Eosin (H&E)–Masson tricrome for inflammation, fat and fibrosis in the liver. *Conclusions*: In this study, HSE was found to slow the development of liver damage in experimental animals that were given a vitamin B12-deficient diet.

## 1. Introduction

Non-alcoholic fatty liver disease (NAFLD) is the most common chronic liver disease. Younessi estimates that 30% of the world has NAFLD [1]. Advanced liver disease, cirrhosis, and HCC exacerbate NAFLD [2]. Despite its high prevalence, a worsening of NAFLD prognosis may develop unnoticed. Thus, preventing steatosis in NAFLD requires a molecular knowledge of its pathophysiology. The accumulation of liver fat in hepatocytes causes steatohepatitis due to oxidative stress. Oxidative stress induces the release of profibrogenic inflammatory cytokines, which promote the development of liver fibrosis and possible cirrhosis [3]. Recent studies have shown that nutritional deficiency conditions cause steatosis or steatohepatitis. Sianipar et al. found that a vitamin B12 deficiency causes hepatic steatosis and steatohepatitis [4,5]. The experimental colitis model established by Harb et al. explained how a vitamin B12 deficiency causes liver inflammation as follows: methionine synthase, the enzyme responsible for converting homocysteine to methionine, requires vitamin B12 as a cofactor. A deficiency of vitamin B12 causes hyperhomocysteinemia (Hhcy) [5]. Hyperhomocysteinemia (Hhcy) is a risk factor for heart disease and other inflammatory diseases, including various NAFLDs such as steatosis or steatohepatitis, due to the effects of cell toxicity. Werstuck et al. proved that Hhcy can increase intracellular homocysteine, interfere with endoplasmic reticulum function, and cause endoplasmic reticulum stress [6]. Hyperhomocysteinemia (Hhcy), due to a lack of vitamin B12, occurs most frequently in high-risk groups such as the elderly, people with gastric mucosal disorders, and in diabetics who consume metformin [6,7,8,9]. 

The herbal rosella plant (*Hibiscus sabdariffa*) is a popular traditional medicine among today’s consumers. *Hibiscus sabdariffa* ruber is an annual, erect, bushy, herbaceous subshrub. Its green leaves have reddish veins and long or short petioles. The flowers, borne singly in the leaf axils, are yellow or buff with a rose or maroon eye, and turn pink as they wither. The typically red calyx stems and leaves have a sour taste [10]. Zhang et al.’s meta-analysis on the extract of *rosella* in capsules and tea showed that the lipid profile in patients with metabolic syndrome improved [11]. In rats, calyxs *Rosella aquouse* extract reduced insulin resistance, hyperglycemia, dyslipidemia and oxidative stress caused by high fructose consumption [12]. The majority of existing research employed *rosella* extract as a treatment for metabolic syndrome conditions. Gossypeptin, anthocyanin, glucoside hibiscin and flavonoids are active ingredients in the rosella extract [13]. Roselle calyx extract contains anthocyanins, which are antioxidants. Anthocyanins are a type of flavonoid component that is found in the calyx of roselle plants [7]. According to Kao et al., the flavonoids in roselle calyx are best extracted using ethanol [8]. As a treatment for hyperlipidemia, it appears promising. Still, higher-quality animal and human studies, as well as information based on actual therapeutic practice, are required to make recommendations for its uses, with broad public health implications [6,9]. Several studies found that *rosella* is effective in lowering triglyceride levels in the liver, implying that *rosella* may act as an anti-steatotic agent [14,15]. However, it has not been established whether *rosella* improves the steatosis process in vitamin B12-deficient animals. This study aimed to examine whether the rosella flower extract could prevent the advancement of NAFLD, steatosis and steatohepatitis caused by homocysteine toxicity in experimental rats on a vitamin B12-restricted diet.

## 2. Materials and Methods

*H. sabdariffa*’s determination was carried out at the laboratory of the Center for Biopharmaca Studies, Bogor Agricultural University. The petals of the *H. sabdariffa* flower were chopped into pieces and dried. Simplicia was dried and powdered until it formed. The extract was prepared from around 100.0 g of simplicia powder using the maceration method, with 750 mL of 70% ethanol added before it was poured, covered and left for five days protected from light, with frequent stirring. After five days of being sprinkled with a burner, pulp and enough pollen, the extract was blended and filtered to collect 1000 mL of juice. The juice was then covered and stored in a cold place for two days, and protected from light until a thick extract was obtained. Then, an evaporator was used to separate, concentrate and evaporate the precipitate. The extract of ethanol from *H. sabdariffa* (HSE) was prepared by PT. Biofarmaca (Bogor, Indonesia). A phytochemical analysis was performed to evaluate the bioactive compounds in the rosella extract. Using the microscopy–microchemical method, the results showed that flavonoid compounds, tannins and saponins were present.

A total of 30 male Sprague–Dawley rats, aged 35 weeks, were ordered from the laboratory of the National Institute of Health Research and Development (Litbangkes) of the Indonesian Ministry of Health in Jakarta. The Health Research Ethics Committee of the Faculty of Medicine, University of Indonesia/Cipto Mangunkusumo Hospital (FKUI/RSCM) number: KET-540/UN2. F1/ETIK/PPM.00.02/2020 (approval date: 3 June 2020), approved the trial procedure in accordance with the ARRIVE guidelines for this study. Prior to conducting the experiment, the animals were acclimatized and handled under controlled conditions for two weeks. The number of experimental animals was 30, according to the Federer formula. The experimental animals were divided into six groups at random, with the treatment time distribution occurring between the ages of eight and sixteen weeks. The control group consisted of a normal diet without the roselle ethanol extract (HSE) (*n* = 5), a vitamin B12-restricted diet group without the HSE (*n =* 5), and a vitamin B12-restricted diet group with the HSE (*n =* 5); each group had two treatment times, which were eight weeks and sixteen weeks, respectively. The composition of the modified vitamin B12- restricted diet from the AIN 93 m diet contained 3 mcg/kg of vitamin B12 (5G1R, Mod TestDiet, Richmond, IN, USA), while the control diet contained 27 mcg/kg of vitamin B12 (58M1, Maintenance TestDiet, Richmond, IN, USA). According to their age at treatment, the experimental animals in the treatment group received HSE intragastrically every day; the group without the HSE was given the same volume of distilled water as HSE with the same route. The dose of the rosella extract in this study was 400 mg/kg BW. This dose was administered according to a study by Andraini and Yolanda in 2015, using the *Hibiscus sabdariffa* extract at a dose of 400 mg/kg/weight/day [16,17]. Their food intake was monitored daily, and theirweight was measured weekly. After 8 and 16 weeks, rats were fasted for 12 h and then euthanized; their blood and liver were collected for experimental testing, and their body weight was recorded [18]. Rat livers were dissected for histological analysis and stored at 80 °C until required for biochemical analysis.

Blood samples were collected at 8 and 16 weeks of the feeding period. Serum samples were covered using centrifugation (3000 rpm for 15 min at 4 °C). Vitamin B12 (MyBioSource, San Diego, CA, USA) and the homocysteine level (MyBioSource, San Diego, CA, USA) in the serum were measured using an ELISA kit, following the manufacturer’s protocol.

After the rats were euthanized, their blood was collected and centrifuged at 3000 rpm for 15 min at 4 °C. Then, the supernatant was collected, and the serum AST (aspartate aminotransferase), ALT (alanine aminotransferase) and GGT (gamma glutamyltransferase) enzyme activity were measured using kits (Randox, Crumlin, UK).

The liver was embedded in paraffin after being immersed in a 10% neutral-buffered formalin. The liver samples were cut into 5 m sections. These findings were examined under a light microscope and categorized as the NAFLD activity score (NAS) based on three characteristics: hepatic steatosis (03), lobular inflammation (03) and hepatic ballooning (03). Hepatic steatosis can be defined as the accumulation of 5% of lipid droplets in the hepatocytes. Lobular inflammation in the liver can be defined as inflammatory cell infiltration and aggregation. Hepatic ballooning refers to swollen hepatocytes and rarefied cytoplasm. The NASs were examined for five different areas per slide, with two slides performed per rat and five rats per group.

The liver was placed in a 10% neutral-buffered formalin embedded in paraffin. The liver samples were cut into 5 m. The results were observed under a light microscope and evaluated for collagen fibers and fibroblasts by observing the blue staining of the preparation. The histological stage of liver fibrosis was measured according to the Metavir Stage F0 assessment, without fibrosis; stage F1, fibrosis in the portal triad; Stage F2, central vein fibrosis.

The homogenized liver samples were analyzed using the Luminex commercial kit, according to the manufacturer’s protocols for the determination of interleukin-6 (IL-6), interleukin-10 (IL-10) and tumor necrosis factor-α (TNF-α) (Millipex, Merck, Germany). The liver inflammation levels were normalized on the basis of the individual protein concentration.

The liver samples were lysed in a RIPA buffer (Sigma-Aldrich, MO, USA). The total protein concentration was determined with a BCA protein assay kit. The processed samples were loaded and run on 12% SDS–polyacrylamide gel before being transferred to PVDF membranes (Merck, MA, USA). The membranes were blocked with 5% skimmed milk. Next, the membranes were incubated at 4 °C overnight with different primary antibodies: SREBP1c (1:500, Abcam, Cambridge, UK), NFkB (1:500, Santa Cruz Biotechnol., Dallas, TX, USA), GRP78 (1:500, Santa Cruz Biotechnol., Dallas, TX, USA) and GADPH (1:1000, Thermo Fisher Scientific, Inc., Waltham, MA, USA). The membranes were washed with TBS- 0.1% Tween 20, and were incubated with fluorescence-conjugated secondary antibodies for 1 h (IRDye 800-680, LiCor, Lincoln, NE, USA). Afterward, the membranes were washed with TBS-0.1% Tween 20. The detection of protein bands was performed using fluorescence detection (Bio-Rad Laboratories Inc., Hercules, CA, USA). The band density from the target protein and the housekeeping protein/loading control (GADPH) were quantified for each batch using Image Lab Software 6.0.1 (Bio-Rad). The ratio of the target protein to GADPH was then calculated, yielding the normalized density of the loading control (NDL); then, the fold difference was calculated by dividing the NDL from each lane by the NDL from the control sample in that batch. This ratio yielded a difference in the sample load between the reference and other lanes. All of the experiments were performed in duplicate to represent the mean ± SD of each rat in all of the data (*n =* 5 animals/group).

The data are presented as the mean ± standard deviation (SD) with 5–7 animals per group (replications are indicated in the figure legend of each experiment). An analysis of variance (One-way ANOVA) was used to detect differences. Tukey’s test was used to correct for multiple comparisons. A value of *p* < 0.05 was considered statistically significant. The data were analyzed using IBM SPSS Statistics version 20.0 (IBM, Armonk, New York, NY, USA).

## 3. Results

### 3.1. Phytochemical Analysis

A phytochemical analysis was performed to evaluate the bioactive compounds in the rosella extract. Using the microscopy–microchemical method, the results showed that flavonoid compounds, tannins, and saponins were present (Table 1).

Results of the phytochemical screening showed the presence of flavonoids, tannins, saponins, quinones and triterpenoids in Table 1. In addition, the total phenol yield using the spectrophotometric method was 3.36% *w*/*w*.

### 3.2. Roselle Extract Reduces Homocysteine Production Due to Vitamin B12 Deficiency

Table 2 shows how the vitamin B12 levels decreased in the group with a B12-restricted diet compared with the control group at both eight and sixteen weeks of age. However, administration of the rosella extract significantly increased vitamin B12 levels compared to the group that was not given the rosella extract. Meanwhile, homocysteine levels increased significantly in the vitamin B12-restricted group compared with the control group. A total of 400 mg/kg BW dose of rosella extract significantly reduced the serum homocysteine levels. This indicates that the rosella extract cut the production of homocysteine caused by limited vitamin B12.

### 3.3. Rosella Extract Protects Liver against Pro-Inflammatory Cytokines Induced by Vitamin B12 Restriction Diet

The dietary restriction of vitamin B12 significantly increased the production of interleukin-6 (IL-6) and tumor necrosis factor-α (TNF-α) compared with the control group. The levels of pro-inflammatory cytokines were significantly reduced after the administration of the rosella extract compared with the vitamin B12-restricted group. Meanwhile, anti-inflammatory cytokine levels significantly increased after rosella extract administration (Table 3). As a result, the rosella extract reduced pro-inflammatory cytokines and increased anti-inflammatory cytokines in the rat liver.

### 3.4. Rosella Extract Alleviated Vitamin B12 Restriction Diet-Induced Liver Histological Changes

A total of 30 hepatic tissue samples from both groups were analyzed with hematoxylin and eosin (H&E)–Masson trichrome staining. In Figure 1, H&E staining revealed steatosis and inflammation in the groups. With H&E staining, a significant increase was observed in the vitamin B12-restricted group compared with the control group, indicating that the vitamin B12-restricted group caused steatosis and steatohepatitis in rats. At eight and sixteen weeks, the rosella extract treatment reduced the features of steatosis and steatohepatitis in H&E staining compared with the vitamin B12-restricted group. Histological studies of the liver tissue with Masson’s trichrome staining at week 16 from the experimental animals showed collagen bands in the vitamin B12-restricted group compared with the control group, and in the vitamin B12-restricted diet with the HSE group (Figure 1).

#### Control Group Vit.B12 Restriction Group Vit B12 Restriction + HSE Group

Table 4 shows the effect of the roselle extract on histological features. The presence of activated resident immune cells, perivascular fibrosis and varying degrees of steatosis indicate the existence of fatty and inflamed liver cells in rats on a vitamin B12-restricted diet group. This procedure began in the eighth week of treatment and progressed until the sixteenth. The HSE treatment helped prevent the development of these conditions by decreasing inflammatory cell activation, fibrosis and steatosis (Table 4).

### 3.5. Roselle Extract Prevents Dietary Vitamin B12 Deficiency-Induced Hepatic Steatosis and Steatohepatitis

The GRP-78 protein expression ratio analysis utilized the Western blot technique, with GAPDH as the reference gene, as shown in Figure 2. When compared with the control group, the dietary limitation in vitamin B12 resulted in a substantial increase in the protein expression ratio of GRP-78. HSE 400 mg/kg BW administration after a vitamin B12-restricted diet resulted in a substantial increase in the GRP-78 protein expression compared with the control group.

As depicted in Figure 2, the protein expression of SREBP1c was observed to be increased in the vitamin B12-restricted group compared to the control group. The administration of HSE 400 mg/kg BW with a vitamin B12-restricted diet resulted in a decrease in SREBP1c protein expression compared with the vitamin B12-restricted diet group without HSE. When compared with the control group, the dietary vitamin B12-restricted group had a significant increase in NFkB protein expression. The administration of HSE 400 mg/kg BW in the vitamin B12-restricted group caused a decrease in the expression of the NFkB protein compared with the vitamin B12-restricted group without the HSE.

## 4. Discussion

For the authors, non-alcoholic fatty liver disease (NAFLD) can be defined as the presence of liver steatosis in patients who consume <20 mg of alcohol per day, as determined via imaging or histology [19]. Pathophysiological occurrences arising from intracellular homocysteine molecules are one of the factors that can cause NAFLD [20]. Increased homocysteine can be caused by a vitamin B12 deficiency; vitamin B12 acts as an enzyme cofactor for methionine synthase [21]. Increased plasma homocysteine levels cause oxidative stress, a lack of methyl donors, misfolded proteins and hepatic reticulum stress [5]. This study discovered that a vitamin B12 deficiency can cause steatosis and inflammation in the liver, with one of the causes being an increase in homocysteine, which is toxic to cells. This was demonstrated by a significant increase in plasma homocysteine levels in rats with a vitamin B12 deficiency. This study supports the findings of Sianipar et al., who discovered an increase in plasma and liver homocysteine in experimental animals with a vitamin B12-restricted diet [3]. The rosella extract was shown to significantly increase plasma vitamin B12 levels while decreasing plasma homocysteine levels. According to Souirti et al., rosella can increase vitamin B12 absorption in the intestines of experimental animals [22]. This is consistent with the findings of our study, wherein plasma vitamin B12 levels increased in experimental animals given a rosella extract. The rosella extract has primarily been shown to reduce homocysteine levels and their toxic effects. The increased absorption of vitamin B12 improves the activity of the methionine synthase enzyme, resulting in lower homocysteine formation.

Liver steatosis is characterized by the accumulation of lipid droplets in hepatocytes. In this study, increased homocysteine was identified as a key pathway for cellular damage to the liver. Moreover, according to the histopathological staining of liver tissue, this increase is strongly linked to structural damage in the liver caused by steatosis, inflammation or fibrosis. According to Ai et al., there is a link between increased homocysteine and the occurrence of steatosis in experimental animals under a high methionine diet [20]. Similarly, Sim et al. discovered an increase in SREBP1c protein expression in the process of rat steatosis with an increase in Hcy due to ethanol induction [23]. Our results agree with their findings, wherein the expression of the SREBP-1c protein increased in the vitamin B12-restricted group, which also experienced an increase in homocysteine compared with the control group. This increase in the SREBP1c protein was associated with a process in the liver known as “de novo lipogenesis.” De novo lipogenesis is a major metabolic pathway in the liver and involves transcription factor SREBP1c, which is responsible for the expression of enzymes in lipogenic processes. We found that rats with a vitamin B12-restricted diet for 16 weeks showed an improvement in features of hepatic steatosis with the rosella extract. Furthermore, the rosella extract was proven to contain phytochemical agents, including flavonoids, quinolones and triacetin [24], compounds which can have health-promoting activities such as antioxidant and anti-inflammatory effects. Our results showed that roselle reduced hepatic fat gain by inhibiting the expression of the transcription factor SREBP1c, resulting in a reduction in the synthesis of new lipids in the liver. This study also supports the findings of Arteaga et al., who studied experimental animals under a high-fat diet. Arteaga’s study demonstrated that the administration of the rosella extract treated attenuated liver steatosis and down-regulated SREBP-1c, resulting in an improvement in steatosis in experimental animals [25]. Chang et al. conducted a clinical trial that demonstrated the effect of rosella extract on reducing obesity, abdominal fat, serum FFA and liver steatosis [15]. Therefore, roselle extract supplementation could help prevent hepatic steatosis caused by limited dietary vitamin B12.

We previously demonstrated that the dietary restriction of vitamin B12 causes oxidative stress, which is a state of cellular imbalance between free radicals and antioxidants, via the formation of homocysteine. The amino acid homocysteine, as previously studied, can cause oxidative stress by increasing the production of hydroxyl radicals [18]. Fats, proteins, carbohydrates and nucleic acids can all be oxidized by this radical group. This pathway’s intermediates can interact with oxygen, resulting in the accumulation of ROS, which can damage cells. In this study, liver damage was caused by limited vitamin B12 in the diet, as evidenced by increased plasma levels of AST, ALT and GGT. The presence of this enzyme was used to detect abnormal liver function. This study found that serum AST and ALT levels were higher in the vitamin B12-restricted group, whereas the roselle extract reduced enzyme levels. A high-fat diet raises serum GGT levels. This study supports the findings of Carvalho et al., who found that plasma homocysteine was higher in NAFLD patients than in controls in their study of 35 patients diagnosed with NAFLD (by liver biopsy) and 45 healthy controls [26]. This was followed by an increase in liver damage enzyme markers such as AST, ALT and GGT. The increase in GGT was more likely caused by an increase in GGT activity, since this enzyme can detect emerging damaged liver cells. The administration of rosella extract has been shown to reduce the activity of enzymes that are markers of liver cell damage. The activity of liver function enzymes, both AST and ALT, was found to be lower in the rosella group compared with the vitamin B12-restricted group who did not receive the rosella extract. Rosella extract contains an active component that has antioxidant activity against ROS. These findings agree with those of Lee et al., who demonstrated that rosella extract could repair liver damage caused by acetaminophen in experimental animals [27].

Generally, the action of antioxidants neutralizes this process. Nonenzymatic (GSH, thioredoxin, vitamins E and C, flavonoids and polyphenols) and enzymatic (SOD, catalase, GPx and peroxyredoxin) antioxidant systems were found to be effective against ROS. NRF2 is a transcription factor that controls the expression of genes that code for antioxidants and detoxifying enzymes. Under physiological conditions, oxidative stress causes endogenous antioxidants and cytoprotective proteins to be up-regulated, hence preventing or limiting tissue damage. The activation of NRF2 mediates this process, increasing the transcription rate of various antioxidant genes and detoxification enzymes [28]. Our results showed that the roselle extracts improved the antioxidant defense against oxidative stress via the NRF2 signaling pathway. Previous research has demonstrated that rosella extract in the liver of vitamin B12-treated rats reduces lipid peroxidation production, and increases levels of antioxidant enzymes such as SOD [18]. In this study, the vitamin B12-restricted group supplemented with rosella extract expressed higher liver NRF2 levels than the group without rosella extract. This suggests that rosella extract maintains oxidant–antioxidant homeostasis in NAFLD rats by increasing antioxidant enzyme defense mechanisms and decreasing lipid peroxidation. Prasomtong et al.’s study on experimental animals under a high-fat diet obtained this effect through the activation of the NRF2 pathway [29].

Furthermore, the activation of ROS caused by vitamin B12 deficiency can be linked to pro-inflammatory processes that activate the NF-kB signaling pathway. NFkB is a transcription factor that regulates many genes, including those involved in inflammatory and proliferative responses. NFkB regulates inflammatory protein and gene expression by inducing the transcription of proinflammatory proteins or genes in response to cellular stimuli, such as excessive ROS production [30]. The results of this study revealed a decrease in NFkB protein expression in the rosella extract group compared with the non-rosella extract group. Arteaga et al. also showed that rosella extract’s content had an anti-inflammatory effect by stopping the NFkB gene from moving or being expressed [25]. Rosella extract functions as an antioxidant that inhibits the phosphorylation process of IκB so that NFkB remains bound to IκB in the cytoplasm, and does not move to the cell nucleus; therefore, NFκB remains passive [6]. According to our findings, a low-vitamin B12 diet raises proinflammatory cytokines similar to TNF-a and IL-6, activating Kupffer cells (liver macrophages) and causing liver inflammation, whereas rosella extract treatment lowers proinflammatory cytokines. This suggests that roselle extract inhibits the development of NAFLD by reducing the release of proinflammatory cytokines. This agrees with our results, which proved that levels of IL-10, as anti-inflammatory cytokines in the liver, increased in the group that was administered the rosella extract. Our results agree with the findings of Lubis et al., who discovered that the expressions of IL-6, TNF- and IL-10 in colitis mice were higher when compared to the controls [31]. Rosella extract contains bioactives that act as antioxidants. These bioactives can capture ROS and free radicals, reduce reactive O2, metabolize fat peroxidation into non-radical products and prevent the generation of free radicals. When ROS levels, TNF-a and IL-6 levels decrease, NFkB activation also decreases [32]. Furthermore, roselle extract increases the production of anti-inflammatory cytokines. TNF-a production is suppressed by increased IL-10 synthesis. Nurkhasanah found that the roselle extract increased the production of anti-inflammatory cytokines (IL-10), which is incongruent with our results [33].

Glucose-regulated protein 78 (GRP78) is a molecular chaperone in the ER that acts as a marker for ER activation. Inducing ER Stress (ERS) in liver cells allows them to mount a resistant and adaptive response to prevent damage via unfolded protein response pathways. If the ERS is prolonged and severe, it can cause apoptosis, inflammation and hepatocyte injury. Zhou et al. discovered that GRP78 levels were elevated in animal models of steatohepatitis [34]. These findings agree with the findings of our current study, which found an increase in GRP78 protein expression in the restricted group compared with the control group. Meanwhile, the administration of roselle extract had no effect on the expression of the GRP78 protein. This could be due to GRP78 undergoing an adaptive process to maintain liver homeostasis. Ye et al.’s experiment on animals fed with a high-fat diet demonstrated that increasing GRP78, which is associated with ER homeostasis, could activate adaptive UPR, which may contribute to its improvement [35]. In experimental animals, the rosella extract prevents steatosis and steatohepatitis due to a vitamin B12 deficiency.

## 5. Conclusions

These studies showed that rosella flower extract reduced the progression of NAFLD in experimental rats that were fed a vitamin B12-restricted diet. The vitamin B12 deficiency led to an increase in plasma homocysteine levels, which caused steatosis and steatohepatitis. The active component of the rosella flower extract inhibited a lipogenesis process in liver cells by lowering the expression of the transcription factor protein SREBP1c. Furthermore, rosella can lower the expression of the NFkB protein, which is responsible for the development of various pro-inflammatory proteins in the liver; it can be assumed that roselle extract suppresses the inflammatory process in the liver through this pathway.

## Figures and Tables

**Figure 1 medicina-59-01044-f001:**
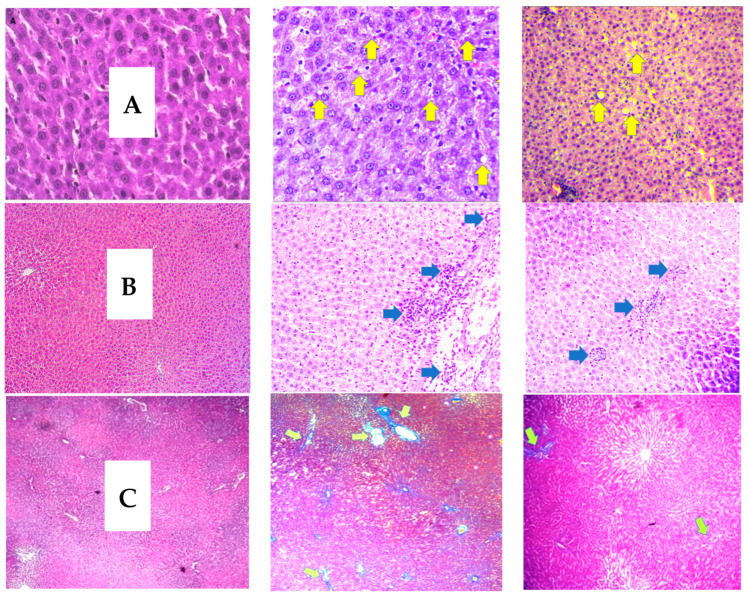
Effect of rosella extract on liver steatosis, steatohepatitis and fibrosis in rats after 16 weeks. Rats were fed a normal diet (control group), a dietary restriction in vitamin B12 only (restriction diet group), and a dietary restriction in vitamin B12 + 400 mg/kg BW roselle extract (Vit B12 restriction diet + HSE). Liver sections were stained with hematoxylin and eosin (H&E) staining (Objective 40×). Enlarged areas of the liver with H&E staining; (**A**) steatosis (yellow arrow) (**B**) lobular inflammation (green arrow) (**C**) liver Masson trichrome staining. Histological stage of liver fibrosis. Stage F0 with no fibrosis, Stage F1 fibrosis in the portal triad, and Stage F2 fibrosis in the central vein. (Objective 20×).

**Figure 2 medicina-59-01044-f002:**
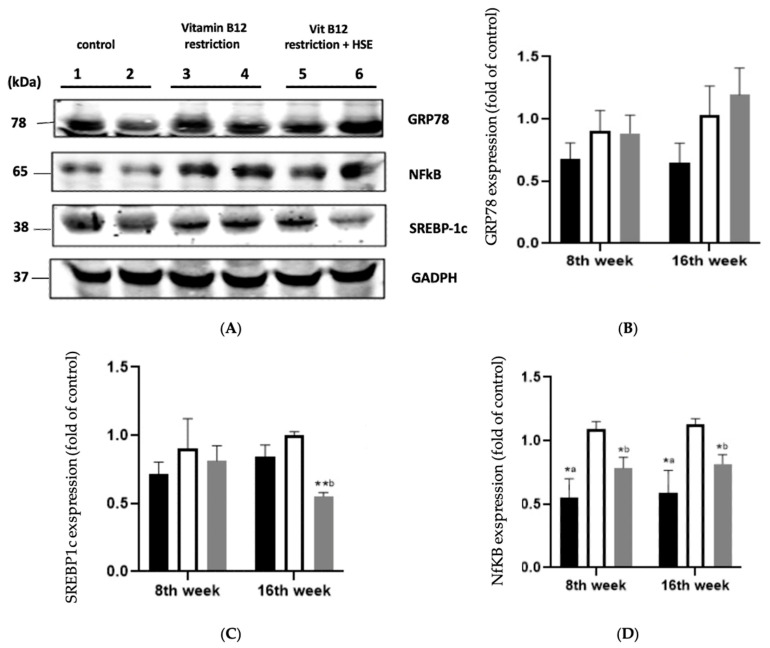
Western blot analysis of GRP78, NFkB and SREBP1c in the rat liver tissue. (**A**) GRP78, NFkB and SREBP1c protein bands in the liver tissue of the control group after 8 weeks (1) and 16 weeks (2); the restriction group after 8 weeks (3) and 16 weeks (4); and the treatment group after 8 weeks (5) and 16 weeks (6). The control group was given a normal diet without HSE, the restriction group was given a vitamin B12-restricted diet without HSE, and the treatment group was given a vitamin B12-restricted diet and HSE at 400 mg/kg weight/day. GADPH was used as an internal control. (**B**) GRP78-relative protein quantification analysis; (**C**) NFkB analysis relative to protein quantification; and (**D**) SREBP1c-relative protein quantification analysis. Bars represent different treatment ages for each group of experimental animals. HSE = *Hibiscus sabdariffa* ethanol extract, GRP78 = glucose-regulated protein-78, NFkB = nuclear factor kappa B, SREBP-1c = sterol regulatory element-binding protein 1c, GADPH = glyceraldehyde 3-phosphate dehydrogenase. Black box: control group; white box: Vit.B12 restriction group; grey box: Vit B12 restriction + HSE. Data are presented as the mean ± SD. Mean differences were calculated by one-way ANOVA followed by Tukey’s post hoc test. The mean difference was significant between control group and restriction group (*a *p* < 0.05) and the treatment group compared to the restriction group (*b *p* < 0.05, **b *p* < 0.01).

**Table 1 medicina-59-01044-t001:** The phytochemical analysis of roselle flower extract.

Metabolite	Results	Analytical Method
Flavonoids	positive	microscopy–microchemical
Tannins	positive	microscopy–microchemical
Saponins	positive	microscopy–microchemical
Quinone	positive	microscopy–microchemical
Triterpenoids	positive	microscopy–microchemical
Total phenol	3.36% *w*/*w*	spectrophotometric

**Table 2 medicina-59-01044-t002:** The effect of roselle extract on animal and biochemical parameters.

Parameter (Plasma)	Feeding Duration (Week)	Control	Vit B12-Restricted Diet	Vit B12-Restricted Diet + HSE
Vitamin B12 (pg/mL)	8	213 ± 6.75	176 ± 5.39 **^,###^	233 ± 10.8 ^+++^
	16	231 ± 4.80	190 ± 1.88 ***^,###^	226 ± 3.89 ^+++^
Homocysteine (pg/mL)	8	1.86 ± 0.07	2.63 ± 0.10 ***^,##^	2.25 ± 0.08 **^,++^
	16	1.92 ± 0.04	2.64 ± 0.07 ***^,###^	2.18 ± 0.09 *^,+++^
ALT activity (U/L)	8	19.8 ± 0.79	27.9 ± 1.16 *	25.4 ± 2.86
	16	20.5 ± 0.89	30.3 ± 1.86 *	27.9 ± 1.43
AST activity (U/L)	8	76.8 ± 6.87	83.2 ± 14.6	77.4 ± 9.06
	16	60.9 ± 8.98	83.1 ± 6.93	65.3 ± 6.09
GGT Activity (U/L)	8	15.4 ± 1.73	22.9 ± 0.10 **^,#^	16.3 ± 0.88 ^+^
	16	17.7 ± 1.31	28.1 ± 3.22 **^,#^	19.3 ± 1.40 ^+^

AST = Aspartate Aminotransferase, ALT = Alanine Aminotransferase, GGT = Gamma Glutamyltransferase. Data are presented as mean ± SD values (*n =* 5 per group). The statistical differences were revealed by one-way ANOVA and Tukey’s post hoc test. The mean differences were significant compared with the control group (* *p* < 0.05, ** *p* < 0.01, *** *p* < 0.001), restriction group (+ *p* < 0.05, ++ *p* < 0.01, +++ *p* < 0.001) and restriction + HSE group (# *p* < 0.05, ## *p* < 0.01, ### *p* < 0.001).

**Table 3 medicina-59-01044-t003:** The effect of roselle extract on inflammatory and anti-inflammatory levels.

Parameter (Liver Homogenate)	16 Weeks		
	Control	Vit B12-Restricted Diet	Vit B12-Restricted Diet + HSE
IL-6 (ng/mg protein)	11.6 ± 0.42	18.5 ± 1.88 **^,#^	14.1 ± 0.51 ^+^
TNF-α (ng/mg protein)	4.58 ± 0.19	7.22 ± 0.34 ***^,##^	5.21 ± 0.29 ^++^
IL-10 (ng/mg protein)	2.47 ± 0.14	1.93 ± 0.06 **	1.92 ± 0.06
NRF2 (pg/mg protein)	225.4 ± 6.31	153.2 ± 17.4 **^,###^	364.4 ± 11.1 ***^,+++^

IL-6 = Interleukin-6, TNF-α = Tumor Necrosis Factor alpha, NRF2 = Nuclear factor-erythroid-2 Related Factor 2, IL-10 = Interleukin 10. Data are presented as mean ± SD values (*n =* 5 per group). The statistical differences were revealed via one-way ANOVA and Tukey’s post hoc test. The mean differences were significant compared with the control group (** *p* < 0.01, *** *p* < 0.001), restriction group (+ *p* < 0.05, ++ *p* < 0.01, +++ *p* < 0.001) and restriction + HSE group (# *p* < 0.05, ## *p* < 0.01, ### *p* < 0.001).

**Table 4 medicina-59-01044-t004:** The effect of roselle extract on histological features.

Parameter	Feeding Duration (Week)	Control	Vit B12 Restriction Diet	Vit B12 Restriction Diet + HSE
Steatosis G1:G2:G3:G4	8	0 0 0 0 0	1 3 1 1 3	1 0 1 1 1
	16	1 0 0 0 1	1 3 3 2 1	1 2 1 1 0
Fibrosis stage F0:F1:F2:F3:F4	8	F0 F0 F0 F0 F0	F0 F1 F1 F2 F2	F0 F0 F1 F1 F1
	16	F0 F1 F0 F0 F1	F2 F2 F2 F2 F1	F1 F2 F1 F1 F1
Inflammation score 0:1:2:3	8	0 1 0 0 1	3 2 1 2 1	1 2 1 1 3
	16	1 1 0 0 1	3 3 3 2 3	1 3 3 2 1

Degree of steatosis: G1 = Mild Steatosis, G2 = Moderate Steatosis, G3 = Severe Steatosis, G3 = Very Severe Steatosis. Fibrosis stage: F0 = absent, F1 = mild/moderate fibrosis, F2 = perisinusoidal and periportal fibrosis, F3 = connecting cirrhosis, F4 = cirrhosis. Inflammation score: 0 = no focus; 1 ≤ 2 focuses per 200× field; 2 = 2–4 focuses per 200× field; 3 ≥ 4 foci per 200× field.

## Data Availability

All data can be found in the manuscript.

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
