# Peer review of "Effect of Roselle Flower Extract (Hibiscus sabdariffa Linn.) on Reducing Steatosis and Steatohepatitis in Vitamin B12 Deficiency Rat Model"

_medicina, 2023, doi:10.3390/medicina59061044_

Round 1

Reviewer 1 Report

Although the primary objective of the study is interesting and could provide novel results on the potential role of roselle flower extract on preventing steatosis progression, there are a lot of important flaws that make impossible to properly review this manuscript. This is mainly associated to lacking results that authors have not included, both in figures and tables:

       Abstract is too long. Please, check author guidelines and adjust the abstract so readers can understand the main content of the manuscript.

          Check thoroughly the abbreviations in the abstract and main manuscript. There are several abbreviations that are not defined in the first use.

      In the introduction section, authors broadly described the main reasons for performing a vitamin B12-deficient mouse model of steatosis. However, this could be summarized from two paragraphs (lines 54-96) to only one shorter.

             In lines 131-133 it is described that the microscope-microchemical method was used to identify flavonoids compounds. Do the authors have images to show these results and include them in the article? It would be very interesting if authors could provide these results as part of the study.

                 Contradictory information is provided regarding the animal experiments. Authors stated that all rats are 35-week-old at the beginning, however, in the description of each groups, they stated that rats are 8-week or 16-week old. After reading the whole article, I could guess that 8 and 16 weeks are the experiment durations for the corresponding groups. Please, clarify this issue in the methodology, since providing reliable and clearer information is mandatory. Moreover, in the abstract, it is described that 35 rats were included in the study and divided into 3 groups.

           Body weight of rats was monitored during the mouse model experiment. However, these results are not included in the article. This information could benefit future research in this field; thus, authors should include it.

          The vitamin B12 and homocysteine serum levels were also measured, together with the parameters ALT, AST and GGT. Please, include also these results in the article.

             Authors should describe in detail how they measured inflammatory markers (IL-10, etc.). They reported that these molecules were determined in the liver samples, however, authors did not describe if mRNA or protein levels were measured.

            Please provide results from the subsection 3.1. Authors cannot describe results that are not showed.

            –  There are not Table 1, neither referenced nor included in the manuscript.

             –  Table 2 is not included in the manuscript.

                In Table 3 authors used the same codes for footnotes as for significance.

              In Figure 1 some immunohistochemical panels have been flipped (the second one has “B” flipped). Moreover, there are two “C” panels, and there are not a logical labeling and description of the panels included in this figure.

                 The same issue described in the previous bullet is observed in Figure 2.

             In Figure 3, statistical results for GRP78 levels analysis are not included in the graph. In addition, statistics of SREBP1c and NfkB determination is confusing, there are “a”, “b” and “*” in the graphs and not defined in the figure legend, while in the figure legend there are “+” and “#” that are not showed in the graphs.

Extensive English editing is needed.

Check carefully the manuscript, some mistakes have been observed (line 117: “stetosis”).

Author Response

Dear Reviewer,

Thank you for taking the time to read and provide feedback on my manuscript. Your input is greatly appreciated and has been invaluable in improving the quality of my work.

Based on your suggestions, I have made several revisions to the manuscript, including clarifying certain points and expanding upon others. I have also addressed the issues you raised regarding the pacing and structure of the piece, and believe that these changes have made the overall flow of the work much smoother.

I have also taken your comments on board regarding the tone and language used throughout the manuscript, and have made adjustments to ensure that the narrative remains engaging and accessible to readers.

Once again, thank you for your thoughtful and constructive feedback. I feel that the manuscript is much stronger as a result of your insights, and I look forward to hearing your thoughts on the revised version. Please see the attachment

Best regards,

[Irena Ujianti]

Reviewer 2 Report

This manuscript describes the effect of Roselle Flower Extract (Hibiscus sabdariffa Linn.) on steatosis and steatohepatitis in the Vitamin B12 Deficiency Rat Model. Overall, this topic is original and demonstrates that this traditional medicine may produce anti-nafld effects in this model. However, there are still questions that need to be addressed.

1.There are problems with the formatting of the full text. The abstract is difficult to read because it is not structured. References are missing and it is impossible to determine if the appropriate literature is cited.

2.How safe this extract is in this animal model needs further elucidation.

3.The language of the whole text needs to be enhanced.

4.There are problems with the layout and numbering of the figures. More figures or tables are needed to clarify the specific benefits of this extract.

A lot of improvement is needed.

Author Response

(The authors gave the same response as above.)

Round 2

Reviewer 1 Report

Authors have deeply and thoroughly revised the article, corrected all the mistakes highlighted and following all the suggestions made. In this new version, a markedly increased quality is observed. Nonetheless, there are still some minor issues that should be corrected:

-     Check thoroughly the abbreviations in the abstract and main manuscript. There are some abbreviations defined that are not used in some cases. For example:

-     In the abstract HSE is defined, but in line 39 “Hibiscuss sabdariffa ethanol extract” is also used.

-     NAFLD is twice defined, in the introduction and again in the discussion section.

-     ROS is not defined.

-        -     Please, cite the tables and/or figures when describing the results presented in them. For example, Table 1 is not mentioned in the subsection 3.1. Therefore, authors should include, as an example: “Using the microscopy-microchemical method, the results showed that flavonoid 185 compounds, tannins, and saponins were present (Table 1).”.

-    -     Figure 2 should be still improved. The color legend for representing the “control group”, “restricted group” and “restricted+HSE group” doesn’t match the color in the graph. Specifically, vitamin B12 restricted group should be represented in the legend as a white box with black edge.

-     Moreover, statistical analysis in this figure is still confusing. In the figure legend, authors describe that a* is significant versus the restriction and control groups, but in the graphs “a*” is placed above the control bars. In the case of “*b” significance, results are clearer presented.

-        -     Discussion section has some minor flaws.

-     Some lines should be referenced. For example, lines 326-328, 365-367.

-        -     NRF2 and Nrf2 are indistinctly used, please be consistent.

-     -     All references are missing.

Author Response

Dear Reviewer,

Thank you for taking the time to review our manuscript entitled Effect of Roselle Flower Extract (Hibiscus sabdariffa Linn.) on Reducing Steatosis and Steatohepatitis in Vitamin B12 Deficiency Rat Model. We appreciate your thoroughness and thoughtful comments, which have helped us to improve the quality and clarity of our work.

We have carefully considered each of your comments and have made the necessary revisions to our manuscript. We have also included a detailed response to each of your comments, which we hope will demonstrate our careful attention to your feedback. Specifically, we have addressed your concerns regarding. We have revised the manuscript accordingly, and we believe that these changes improve the clarity and accuracy of our work.

We hope that you will find our revisions and responses satisfactory, and we appreciate your help in ensuring the quality of our manuscript. We believe that our revised manuscript now meets the high standards of Medicina, and we look forward to your decision regarding its publication.

Thank you again for your valuable feedback, and please do not hesitate to contact us if you have any further questions or concerns.

Sincerely,

Irena Ujianti

Reviewer 2 Report

This version is ok. Reference list is still missing.

Author Response

(The authors gave the same response as above.)

Round 3

Reviewer 1 Report

Regarding last changes made, cover letter has not been attached (or an error in the website impedes me to download it), therefore I cannot see the specific answers of authors. Nonetheless, most issues have been amended, although there is one issue that should be still improved:

-    In the figure legend of Figure 2 it is described that “The mean difference was significant compared to the control and restriction groups (a*P<0.05) and the treatment group compared to the restriction group (b*P<0.05, b**P<0.01).” Statistics of *a are not properly explained. In my opinion the mistake has two alternatives to be corrected:

1.      Authors are correctly explaining the groups to which significance tests were made: *a represents significance compared to control group and restriction group. In this case, *a should be located on the bar graphs of vit B12 restriction+HSE group.

2.      Otherwise, in this sentence authors mean that *a represents significance between control group and restriction group. In this case, *a is correctly placed in the graphs, but the sentence on the figure legend should be corrected.

Please, it is mandatory to describe clearly the results, thus, amend this issue.

-       References included has important mistakes, not only regarding the format, but also the mismatching with the citations. Some examples:

1.      Reference 9: the title of the article is duplicated. Authors list is finished with “et al”, but there are not more authors than these described. Article volume and pages are not included.

2.      Reference 7 is not an online website, but the link has been added.

3.      Mismatching: line 53, “Sianipar et al.” is referenced in “5”, but this reference is “Rodríguez-Ramiro et al.”. Line 55, “Harb et al.” is referenced in “9”, but this reference is “Olatunji et al”.

- A correct matching and use of relevant articles in the field is strictly needed in a scientific paper. Check ALL the citations and references so they are correctly included

Author Response

Dear Reviewer,

Thank you for your insightful comments on our manuscript. We appreciate your feedback and have carefully considered your suggestion regarding the need for a correct matching and use of relevant articles in the field.

We have thoroughly checked all the citations and references in our manuscript and made sure that they are correctly included. We understand the importance of citing relevant articles in the field and have taken extra care to ensure that all the references are accurate and up-to-date.

We have also carefully considered your feedback regarding the legend in our figure and have made the necessary changes. The legend now reads as follows: 'The mean difference was significant between control group and restriction group (a*P<0.05)'.

We hope that our revisions meet your expectations and that you find the new version of our manuscript to be satisfactory. Once again, we thank you for your valuable feedback and appreciate your contributions to the field of research.

Sincerely,

Irena Ujianti

Answer For Reviewer

  1. In the figure legend of Figure 2 it is described that “The mean difference was significant compared to the control and restriction groups (a*P<0.05) and the treatment group compared to the restriction group (b*P<0.05, b**P<0.01).” Statistics of *a are not properly explained. In my opinion the mistake has two alternatives to be corrected:
  2. Authors are correctly explaining the groups to which significance tests were made: *a represents significance compared to control group and restriction group. In this case, *a should be located on the bar graphs of vit B12 restriction+HSE group.
  3. Otherwise, in this sentence authors mean that *a represents significance between control group and restriction group. In this case, *a is correctly placed in the graphs, but the sentence on the figure legend should be corrected. Please, it is mandatory to describe clearly the results, thus, amend this issue.

response: We are very grateful for the directions given by the reviewers, and after careful consideration, we would like to choose the second option as per the reviewers' directions. We have made the necessary changes to the legend and it now reads as follows: 'The mean difference was significant between control group and restriction group (a*P<0.05)'. Once again, we thank you for your insightful comments and suggestions that have helped us to improve our manuscript.

  1. References included has important mistakes, not only regarding the format, but also the mismatching with the citations. Some examples:
    1. Reference 9: the title of the article is duplicated. Authors list is finished with “et al”, but there are not more authors than these described. Article volume and pages are not included.
    2. Reference 7 is not an online website, but the link has been added. 
    3. Mismatching: line 53, “Sianipar et al.” is referenced in “5”, but this reference is “Rodríguez-Ramiro et al.”. Line 55, “Harb et al.” is referenced in “9”, but this reference is “Olatunji et al”. 

 A correct matching and use of relevant articles in the field is strictly needed in a scientific paper. Check ALL the citations and references so they are correctly included

Response: Thank you for your insightful comments on our manuscript. We appreciate your feedback and have carefully considered your suggestion regarding the need for a correct matching and use of relevant articles in the field. We have thoroughly checked all the citations and references in our manuscript and made sure that they are correctly included. We understand the importance of citing relevant articles in the field and have taken extra care to ensure that all the references are accurate and up-to-date. We hope that our revisions meet your expectations and that you find the new version of our manuscript to be satisfactory. Once again, we thank you for your valuable feedback.

  1. Younossi, Z.; Anstee, QM.; Marietti, M.; Hardy, T.; Henry, L.; Eslam, M.; et al. Global burden of NAFLD and NASH: trends, predictions, risk factors and prevention. Nat Rev Gastroenterol Hepatol. 2017, 14, 11–20.
  2. Kanda, T.; Goto, T.; Hirotsu, Y.; Masuzaki, R.; Moriyama, M.; Omata, M. Molecular mechanisms: Connections between nonalcoholic fatty liver disease, steatohepatitis and hepatocellular carcinoma. Int J Mol Sci. 2020, 21, 1525.
  3. Sianipar,, IR.; Ujianti, I.; Yolanda, S.; Murthi, AK.; Amani, P.; Santoso, DIS. Developing vitamin B12 deficient rat model based on duration of restriction diet: Assessment of plasma vitamin B12, homocysteine (Hcy), and blood glucose levels. InAIP Conference Proceedings. 2019, 2092, 020004.
  4. Sianipar, IR.; Ujianti, I.; Yolanda, S.; Yusuf, AA.; Kartinah, NT.; Amani, P. et al. Low vitamin B12 diet increases liver homocysteine levels and leads to liver steatosis in rats. Universa Medicina. 2019, 38, 194–201.
  5. Harb, Z.; Deckert, V.; Bressenot, AM.; Christov, C.; Guéant-Rodriguez, R-M.; Raso, J.; et al. The deficit in folate and vitamin B12 triggers liver macrovesicular steatosis and inflammation in rats with dextran sodium sulfate-induced colitis. J Nutr Biochem. 2020, 84, 108415.
  6. Da-costa-rocha, I.; Bonnlaender, B.; Sievers, H.; Pischel, I.; Heinrich, M. Hibiscus sabdariffa L . – A phytochemical and pharmacological review. Food Chem. 2014, 165, 424–443.
  7. Zhang, B.; Yue, R.; Wang, Y.; Wang, L. Effect of Hibiscus sabdariffa ( Roselle ) supplementation in regulating blood lipids among patients with metabolic syndrome and related disorders : A systematic review and meta-analysis. Phytother Res. 2019, 34, 1083–1095.
  8. Ajiboye, TO.; Raji, HO.; Adeleye, AO.; Oladiji, AT. Hibiscus sabdariffa calyx palliates insulin resistance , hyperglycemia , dyslipidemia and oxidative rout in fructose-induced metabolic syndrome rats. Sci. Food Agric. 2016, 96, 1522-1531.
  9. Ojulari, OV.; Lee, SG.; Nam, J. Beneficial Effects of Natural Bioactive Compounds from Hibiscus sabdariffa L . on Obesity. Molecules. 2019, 24, 210.
  10. Rodriguez-Ramiro, I.; Vauzour, D.; Minihane, AM. Polyphenols and non-alcoholic fatty liver disease: Impact and mechanisms. Proc Nutr Soc. 2016, 75, 47–60.
  11. Kao, E-S.; Yang, M-Y.; Hung, C-H.; Huang, C-N.; Wang, C-J.; Polyphenolic extract from Hibiscus sabdariffa reduces body fat by inhibiting hepatic lipogenesis and preadipocyte adipogenesis. Food Funct. 2016, 7, 171–182.
  12. Guardiola, S.; Mach, N. Therapeutic potential of Hibiscus sabdariffa: A review of the scientific evidence. Endocrinol Nutr.2014, 61, 274–295.
  13. Hopkins, AL.; Lamm, MG.; Funk, JL.; Ritenbaugh, C. Fitoterapia Hibiscus sabdariffa L . in the treatment of hypertension and hyperlipidemia : A comprehensive review of animal and human studies. 2013, 85, 84–94.
  14. Olatunji, LA; Adebayo, JO.; Oguntoye, OB.; Nafisat, O.; Olatunji, VA.; Soladoye, AO.; et al. Effects of Aqueous Extracts of Petals of Red and Green Hibiscus sabdariffa on Plasma Lipid and Hematological Variables in Rats. Biol. 2005, 43, 471-474.
  15. Chang, HC.; Peng, CH.; Yeh, DM.; Kao, ES.; Wang, CJ. Hibiscus sabdariffa extract inhibits obesity and fat accumulation, and improves liver steatosis in humans. Food Funct. 2014, 5, 734–739.
  16. Jamrozik, D.; Borymska, W.; Kaczmarczyk-Å»ebrowska, I. Hibiscus sabdariffa in Diabetes Prevention and Treatment—Does It Work? An Evidence-Based Review. Foods. 2022,11, 2134.
  17. Andraini, T.; Yolanda, S.; Prevention of insulin resistance with Hibiscus sabdariffa Linn. extract in high-fructose fed rat. Med J Indones. 2015, 23, 192.
  18. Ujianti, I.; Sianipar, I.R.; Prijanti, A.R,; Santoso, D.I.S. Consumption Of Hibiscus Sabdariffa Dried Calyx Ethanol Extract Improved Redox Imbalance And Glucose Plasma In Vitamin B12 Restriction Diet In Rats. Appl. Biol. 2022, 51, 33–40.
  19. Younossi, ZM.; Marchesini, G,; Pinto-cortez, H.; Petta S. Epidemiology of Nonalcoholic Fatty Liver Disease and Nonalcoholic Steatohepatitis : Implications for Liver. Transplantation. 2019, 103, 22-27.
  20. Ai, Y.; Sun, Z.; Peng, C.; Liu, L.; Xiao, X.; Li J. Homocysteine induces hepatic steatosis involving ER stress response in high methionine diet-fedmice. Nutrients. 2017, 9, 1–10.
  21. Ueno, A.; Hamano, T.; Enomoto, S.; Shirafuji, N.; Nagata, M.; Kimura H. et al. Influences of Vitamin B12 Supplementation on Cognition and Homocysteine in Patients with Vitamin B12 Deficiency and Cognitive Impairment. Nutrients. 2022, 14, 1–14.
  22. ; Loukili, M.; Soudy, ID.; Rtibi, K.; Özel, A.; Limas-Nzouzi, N. et al. Hibiscus sabdariffa increases hydroxocobalamin oral bioavailability and clinical efficacy in vitamin B12 deficiency with neurological symptoms. Fundam Clin Pharmacol.2016, 30, 568–576.
  23. Sim, WC.; Yin, HQ.; Choi, HS.; Choi, YJ.; Kwak, HC.; Kim, SK. et al. L-serine supplementation attenuates alcoholic fatty liver by enhancing homocysteine metabolism in mice and rats. J Nutr. 2015, 145, 260–267.
  24. Gad, FAM.; Farouk, SM.; Emam, MA. Antiapoptotic and antioxidant capacity of phytochemicals from Roselle (Hibiscus sabdariffa) and their potential effects on monosodium glutamate-induced testicular damage in rat. Environ Sci Pollut Res. 2021, 28, 2379–2390.
  25. Villalpando-Arteaga, EV.; Mendieta-Condado, E.; Esquivel-Solís, H.; Canales-Aguirre, AA.; Gálvez-Gastélum, FJ.; Mateos-Díaz, JC. et al. Hibiscus sabdariffa L. aqueous extract attenuates hepatic steatosis through down-regulation of PPAR-γ and SREBP-1c in diet-induced obese mice. Food Funct. 2013, 4, 618–626.
  26. De Carvalho, SCR.; Muniz, MTC.; Siqueira, MDV.; Siqueira, ERF.; Gomes, AV.; Silva, KA. et al. Plasmatic higher levels of homocysteine in Non-alcoholic fatty liver disease (NAFLD). Nutr J. 2013, 12, 6–10.
  27. Lee, CH.; Kuo, CY.; Wang, CJ.; Wang, CP.; Lee, YR.; Hung, CN. et al. A polyphenol extract of Hibiscus sabdariffa L. Ameliorates acetaminophen-induced hepatic steatosis by attenuating the mitochondrial dysfunction in Vivo and in Vitro. Biosci Biotechnol Biochem. 2012, 76, 646–651.
  28. Dai, X.; Yan, X.; Wintergerst, KA.; Cai, L.; Keller, BB.; Tan, Y. NRF2: Redox and Metabolic Regulator of Stem Cell State and Function. Trends Mol. Med. 2020, 26, 185–200.
  29. Prasomthong, J.; Limpeanchob, N.; Daodee, S.; Chonpathompikunlert, P.; Tunsophon, S. Hibiscus sabdariffa extract improves hepatic steatosis, partially through IRS-1/Akt and NRF2 signaling pathways in rats fed a high fat diet. Sci Rep. 2022, 12, 1–15.
  30. Guo, Y.; Zhang, X.; Zhao, Z.; Lu, H.; Ke, B.; Ye, X. et al. NF-κB/HDAC1/SREBP1c pathway mediates the inflammation signal in progression of hepatic steatosis. Acta Pharm. Sin. B. 2020, 10, 825–836.
  31. Lubis, M.; Siregar, GA.; Bangun, H.; Ilyas, S. The effect of roselle flower petals extract (Hibiscus sabdariffa linn.) on reducing inflammation in dextran sodium sulfateinduced colitis. Glas. 2020, 17, 1–7.
  32. Al-Rasheed, NM.; Fadda, LM.; Al-Rasheed, NM.; Ali, HM.; Yacoub, HI. Down-regulation of NFkB, Bax,TGF-β, Smad-2mRNA expression in the livers of carbon tetrachloride treated rats using different natural antioxidants. Arch. Biol. Technol.2016, 59, 1–10.
  33. The effect of rosella (Hibiscus Sabdariffa L) treatment on IL-10 and IL-14 secretion on dimethylbenz (A) anthracene (DMBA) induced rat. Int. J. Pharm. Pharm. Sci. 2015, 7, 402–404.
  34. Zhou, X.; Han, D.; Yang, X.; Wang, X.; Qiao, A. Glucose regulated protein 78 is potentially an important player in the development of nonalcoholic steatohepatitis. Gene. 2017, 637, 138–144.
  35. Ye, R.; Jung, DY.; Jun, JY.; Li, J.; Luo, S.; Ko, HJ. et al. Grp78 heterozygosity promotes adaptive unfolded protein response and attenuates diet-induced obesity and insulin resistance. Diabetes. 2010, 59, 6–16.
